# Perspectives on Saponins: Food Functionality and Applications

**DOI:** 10.3390/ijms241713538

**Published:** 2023-08-31

**Authors:** Yakindra Prasad Timilsena, Arissara Phosanam, Regine Stockmann

**Affiliations:** 1Commonwealth Scientific and Industrial Research Organisation (CSIRO), Agriculture and Food, Werribee, VIC 3030, Australia; yakindra.timilsena@csiro.au; 2Department of Food Technology and Nutrition, Faculty of Natural Resources and Agro-Industry, Kasetsart University, Chalermphrakiat Sakon Nakhon Province Campus, Sakhon Nakon 47000, Thailand; arissara.p@ku.th

**Keywords:** saponins, emulsifiers, foaming agents, plant extract, food functionality

## Abstract

Saponins are a diverse group of naturally occurring plant secondary metabolites present in a wide range of foods ranging from grains, pulses, and green leaves to sea creatures. They consist of a hydrophilic sugar moiety linked to a lipophilic aglycone, resulting in an amphiphilic nature and unique functional properties. Their amphiphilic structures enable saponins to exhibit surface-active properties, resulting in stable foams and complexes with various molecules. In the context of food applications, saponins are utilized as natural emulsifiers, foaming agents, and stabilizers. They contribute to texture and stability in food products and have potential health benefits, including cholesterol-lowering and anticancer effects. Saponins possess additional bioactivities that make them valuable in the pharmaceutical industry as anti-inflammatory, antimicrobial, antiviral, and antiparasitic agents to name a few. Saponins can demonstrate cytotoxic activity against cancer cell lines and can also act as adjuvants, enhancing the immune response to vaccines. Their ability to form stable complexes with drugs further expands their potential in drug delivery systems. However, challenges such as bitterness, cytotoxicity, and instability under certain conditions need to be addressed for effective utilization of saponins in foods and related applications. In this paper, we have reviewed the chemistry, functionality, and application aspects of saponins from various plant sources, and have summarized the regulatory aspects of the food-based application of quillaja saponins. Further research to explore the full potential of saponins in improving food quality and human health has been suggested. It is expected that this article will be a useful resource for researchers in food, feed, pharmaceuticals, and material science.

## 1. Introduction

The name ‘saponin’ is derived from the Latin word ‘sapo’ meaning soap, and associated with the ability to form a soapy foam when plant extract containing saponins is agitated in water [1]. Saponins are a diverse group of compounds widely distributed in the plant kingdom, which are characterized by their structure, which comprises a lipophilic triterpene or steroid aglycone linked to one or more hydrophilic sugar moieties [2,3]. Their structural diversity is reflected in their physicochemical and biological properties.

In recent years, consumer demand for natural products coupled with their physicochemical (surfactant) properties and mounting evidence on their biological activity (such as anticancer properties, and hemolytic and hypocholesterolemia activity) has led to the emergence of saponins as commercially important compounds with expanding applications in the food, cosmetics, and pharmaceutical industries [3,4].

Although saponins are known for their health benefits and functional attributes, they also come with some limitations. One of these limitations is associated with their ability to interact with other food components to form complexes, with proteins, lipids, minerals such as iron, zinc, and calcium (insoluble saponin–mineral complexes) negatively influencing the absorption of such food components in the body [5]. Some saponins have limited solubility in water, which can pose challenges in their incorporation and distribution within food matrices. This can affect their effectiveness as emulsifiers or stabilizers in certain food systems [6].

Saponins also find applications in pharmaceuticals and other allied industries [7]. Saponins may help lower blood lipids, reduce cancer risks, and slow blood glucose response [8]. A diet rich in saponins has been suggested to inhibit dental decays and the aggregation of platelets. Saponin consumption has also shown promise in the treatment of hypercalciuria in humans, and as an antidote against acute lead poisoning [8]. Saponins are also used as expectorant and antitussive agents [9]. They are utilized in the cosmetic industry for their ability to act as natural emulsifiers, foaming agents, and cleansing agents. They are constituents of shampoos, soaps, facial cleansers, body washes and shaving creams, contributing to the lathering, cleansing, and moisturizing properties of these products [10].

As the functionality and subsequent applications of saponins are dictated by their composition, a thorough understanding of their chemical and structural features is essential. In addition to their plant origin, such as species and variety, the methods of their extraction and purification play a vital role in saponin composition, structure, functionality, and applications. Saponins are highly researched bioactives, and there are several experimental studies and review articles concerning them. When searching databases such as Google Scholar, PubMed, ScienceDirect, SciFinder, Scopus, and the Web of Science, we found over 32,000 articles that describe the different aspects of saponins. For example, a recent review article [11] highlighted their surfactant properties and applications, whereas Cheok et al. [12] described various techniques for the extraction of saponins. Similarly, El Aziz et al. [13] focused on the chemistry, isolation, and methods of quantification of saponins, and the cytotoxic aspects of saponins are well reviewed elsewhere [14]. A recent review describes the potential of saponins from various sources as antiviral agents [15]. However, there is a scarcity of articles that present a concise overview of the sources, methods of extraction, purification, characterization, quantification, and end-use application of saponins. In addition, regulatory requirements in the food and pharmaceutical application of saponins are also not clearly addressed in previous reviews. Therefore, this review paper is expected to fill the gaps in the literature with regard to aspects of saponins’ chemistry, structure, and function, with end-use applications in foods, cosmetics, and pharmaceuticals as a focus.

## 2. Sources of Saponins

Saponins are widely distributed in nature and are found in several plants and some marine animals (e.g., sea stars, sea cucumbers, and sponges) [16]. The major dietary exposure to saponins stems from consumption of legumes. Soybean saponins are one of the common types of legume saponins consumed [3]. Saponin content and their structure and composition may vary even among the same species of edible legumes, because of variations with cultivars as well as growing locations, irrigation conditions, soil types, and climatic conditions [8]. For example, soybean saponins are classified into group A, B, and E based on the chemical structure of the aglycone. Some of the better-known botanicals rich in saponins are presented in Table 1.

Among various botanicals listed in Table 1, liquorice, yucca and quillaja bark are among the plant sources containing the highest quantity of saponins; however, soapwort, milkwort, primula, and fenugreek also contain appreciable quantities of saponins. Sugar beet leaves, Chinese ginseng, and horse chestnuts also contain good quantities of saponins.

## 3. Chemistry of Saponins

Structurally, saponins are a diverse class of compounds with vast functional diversity [24]. As schematically presented in Figure 1 and Figure 2, there are generally two functional groups in the saponin structure: an aglycone (a 30-carbon skeleton molecule) and a glycone (with one or more sugar units) [13]. Aglycone components are made up of a triterpenoid or steroid. In general, the glycone or sugar moiety consists of a monosaccharide, or an oligosaccharide covalently linked to the skeleton molecule at C3 position. In some cases, additional sugars are attached at the C26 or C28 positions [25,26]. The presence of both lipophilic (aglycone) and hydrophilic (sugar) components in one molecule gives them soap-like properties. When saponins are mixed with water or aqueous alcohol and agitated for some time, a rich foam develops, resulting from the diminished surface tension and aggregation into micelles [26].

Broadly, saponins are classified into two main groups based on the structure of the aglycone. They are known as the triterpenoid saponins and steroid saponins. Triterpenoid saponins consist of only 6 rings, with 30 C atoms in total. Steroid saponins contain 5 rings and have only 27 C atoms (Figure 3). The steroid saponins have three methyl groups removed [28]. Some authors recognize a third group, that of the steroidal amines or alkaloid saponins; they have the same basic structure as the steroidal saponins but possess an NH group instead of an O atom [26,28]. The great complexity of the saponin structure arises from the variability of the aglycone structure, the nature of the side chains, and the position of the attachment of these moieties to the aglycone [27].

## 4. Extraction and Isolation of Saponins

The extraction and isolation of saponins from plants is complicated by their structural diversity, low concentrations in plant biomass, and chemical instability. Saponins can be extracted from plant materials using various techniques, as shown in Table 2, and the aim of extraction is to liberate the saponins from the plant cells; maceration of the biomass is an important first step. Simple methods of extraction use either water, alcohols and other solvents as the extraction medium in emersion, percolation, or decoction modes. Soxhlet extraction and the combination of solvent extraction with intensification techniques such as ultrasound or microwave have shown higher efficiencies (e.g., increased yield and productivity) compared with simple solvent extraction [11].

In simple extraction methods, saponins can easily be extracted by immersion or soaking the powdered or macerated biomass in a suitable specific solvent for a specified time period. The polarity of the solvent, temperature, time, mixing speed, solubility of saponins and the effective diffusion in the liquid phase are the main operational variables affecting the efficiency of the extraction process [13]. A longer extraction time, higher temperature, and higher mass of biomass generally result in increased extraction yields.

Water, ethanol, methanol, n-butanol, acetone, ethyl acetate, dichloro methane, or a mixture of solvents are commonly used for the extraction of saponins from plant materials; however, ethanol and n-butanol are more common [13]. The temperature of extraction varies from ambient to the boiling point of the chosen solvent. Le et al. [31] has provided a detailed outline of a method to extract saponins using a mixture of ethanol–water (70% ethanol) at 50 °C.

The Soxhlet extraction process is considered a more efficient process in comparison to the simple extraction process, because during Soxhlet extraction a hot organic solvent directly contacts the plant tissue in the condenser, so the extraction process takes place via direct contact between the plant tissue and the hot fumes of the solvent. After a considerable extraction time, the solvent is rich in the active ingredients [32].

In any solvent extraction process, after the completion of the extraction cycle, the extract is further concentrated using solvent evaporation or vacuum distillation, and can then be dried to a powder. However, after the extraction and concentration steps there may be undesirable impurities present, and further fractionation or purification of saponins may be required [33]. Selection of the purification protocol needs to be designed for the specific targeted saponin molecules to be purified from an extract, and hence only general principles are discussed here.

Extract purification is typically performed after the concentration step. Differential solubility approaches using a series of polar solvents selected on the properties of the saponins and the untargeted impurities may be employed to remove non-saponin impurities from the extracts. Chromatography is a powerful refining technique and can operate in various modes to purify classes of saponins or individual saponins if operated in gradient elution mode. Adsorption of saponins has been demonstrated for various anion and cation exchange resins [34,35]. Non-functionalized macro-porous polystyrene-based resins have also shown binding and elution of saponins, likely facilitated by hydrophobic and pi-stacking interactions between resin surface and saponins. Indeed, there is a polystyrene-based resin commercially available specifically developed for saponin isolation (https://kivifilter.en.made-in-china.com/product/INCEGbrAfOWw/China-Extraction-of-Saponin-Non-Polar-Crosslinked-Polystyrene-Macroporous-Polymeric-Adsorbent.html, accessed on 15 August 2023) [35]. The advantage of process chromatography is that the resin selection and elution conditions can be tailored to make quite high-purity fractions of saponins or individual saponin molecules. Other saponins, like β-escin, the primary active principle of horse chestnut seed and a historical remedy, can be recovered from aqueous extracts via acidification and re-crystallization, with the process being commercially implemented [36]. Dialysis and filtration are often applied as the final polishing step to remove salts and other small molecule contaminants [37].

## 5. Quantification of Saponins

For analytical quantification, extracted saponins are further concentrated, purified, separated, identified and characterized using various chromatographic and spectroscopic methods. Among them, high-performance thin-layer chromatography (HPTLC) and high-performance liquid chromatographic (HPLC) have been widely used for the separation and quantification of individual *Sapindus saponins* [38] and soybean saponins [39]. Reversed-phase HPLC methods with C18 columns were suggested to be more effective in separating most of the naturally occurring saponins [40]. Five different fractions of soybean saponins were separated via HPLC using a C18 reversed-phase column (as shown in Figure 4). The mobile phase was 40% aqueous acetonitrile with 0.025% trifluoroacetic acid [39]. In another study, six pairs of saponin diastereomers were isolated from *Y. schidigera* by using HPLC with a C30 column [41]. The absence of chromophores in many saponins prevents their detection using ultraviolet detectors; however, an evaporative light-scattering detector (ELSD) is more effective for detection purposes [40].

Mass spectrometry in combination with chromatography has been more widely used in the characterization of saponins. Ultra-performance liquid chromatography (UPLC) coupled with an exactive mass spectrometer was used for the characterization of the purified saponins from chubak root extract (*Acanthophyllum glandulosum*) [42]. Another study examined saponin profiles in nine distinct legume seeds, such as soybean, adzuki bean, cowpea, common bean, scarlet runner bean, lentil, chickpea, hyacinth bean, and broad bean. This investigation was carried out by employing the method of ultra-performance liquid chromatography coupled with photodiode array detection and electrospray ionization/mass spectrometry (UPLC-PDA-ESI/MS) [43]. Liquid chromatography-mass spectrometry (LC-MS) was used to quantify saponins from *Y. schidigera* [44]. Similarly, HPLC coupled with the tandem mass spectrometry method was used for quantitative analysis of the steroidal saponins in *Y. gloriosa* flower samples [45].

Gas–liquid chromatography has restricted utility because saponins, being relatively large molecules, lack volatility [46]. NMR and GC-MS demand thorough purification of saponins, and in the instance of GC-MS, saponins are required to be converted to their respective sapogenins, followed by the preparation of volatile derivatives. Both the comprehensive purification process and the subsequent derivatization stages are time-intensive and are recognized for generating unintended distortions [47]. Capillary electrophoresis has been used for the quantification of saponins only in some instances, as this method is still under optimization and validation [46].

Cui et al. [48] utilized electrospray ionization multi-stage tandem mass spectrometry (ESI-MS(n)) and liquid chromatography coupled with online mass spectrometry (LC/MS/MS) to characterize saponins in unprocessed extracts from Panax ginseng. The MS(n) data obtained from the [M-H](−) ions of saponins offer valuable insights into the structural aspects of saccharide chains and sapogenins within saponins. Through ESI-MS(n), the identification of non-isomeric and isomeric saponins with varying aglycones becomes efficient within plant extracts. Additionally, LC/MS/MS serves as a complementary analytical approach, which is particularly effective for discerning isomeric saponins. These methodologies collectively serve as potent analytical tools, enabling swift screening and structural elucidation of saponins within plant extracts [48].

Among several spectroscopic methods, UV-Vis spectroscopy is widely used; however, FTIR spectroscopy to measure saponin content has also been reported [49]. Among various methods of saponin quantification using UV-Vis, the vanillin–sulphuric acid method is quicker and more accurate [31]. This method measures absorbance at 560 nm using a spectrophotometer. Aescin dissolved in methanol can be used as standard. The total saponin content of the extract is expressed as mg aescin equivalents (AE) per gram dry weight of the botanical (mg AE/g).

Table 3 lists some of the analytical methods used for the quantification of saponins. A simple comparison of the key features of these methods such as sensitivity, selectivity, and accuracy are summarized.

The most effective method of quantification of saponins depends on various factors, including the specific saponin compounds of interest, the complexity of the sample matrix, the sensitivity required, and the available resources. There is no one-size-fits-all answer, as different methods have their advantages and limitations. However, high-performance liquid chromatography coupled with mass spectrometry (HPLC-MS) is often considered one of the most effective methods for quantifying saponins [57].

## 6. Functional Properties of Saponins

### 6.1. Techno-Functional Characteristics of Saponins

The physical, chemical, and biological properties of saponins are the result of their structural complexity. These properties are not generic; only a few of them are common to all members of this diverse group of molecules. Almost all saponins are known to have a bitter, unpleasant taste and exist in a colorless, amorphous form [58]. Each saponin has different solubility in different solvents. However, most saponins have good solubility in water, methanol, ethanol, and n-butanol [59]. Factors such as solvent temperature, composition and pH play a key role in the solubility and extractability of saponins.

Saponins have a high melting point (generally above 200 °C) and maintain their biological activities even if they are processed at relatively high temperatures in water at 100 °C for several minutes. However, they are not stable in certain acid or alkaline conditions, as the glycosidic bond (bond between the sugar chain and the aglycone) can easily be hydrolyzed in the presence of acids or alkali [59]. The products of hydrolysis include aglycones, prosapogenins, sugar residues, or monosaccharides, depending on the extent and conditions of hydrolysis [4].

It has been reported that the most important factor determining functional activity of saponins is the structure of aglycone part, in particular, the number and the location of functional groups. The activity level is strongly dependent on the number and the structure of the sugar chains. Generally, monodesmosides show much higher activities than bi- or tridesmosides. Sterol affinity is another important consideration that determines the application of saponins [17].

More details about the functional aspects of saponins will be presented in Section 7, along with their application.

### 6.2. Bioactivities of Saponins

Saponins have been reported to exhibit a wide variety of biological activities. They demonstrate efficacy in combating cancer and inflammation, as well as acting as potent antimicrobial agents. Saponins have also shown promising anticancer effects in numerous studies. They can induce apoptosis (programmed cell death) in cancer cells, inhibit tumor cell proliferation, and suppress angiogenesis (the growth of new blood vessels that nourish tumors). Some saponins have been investigated for their potential to prevent or slow the progression of various types of cancer, including breast, lung, prostate and colon cancers [60]. They can inhibit the production of pro-inflammatory cytokines and enzymes, thereby reducing inflammation and alleviating symptoms in conditions like arthritis, inflammatory bowel disease (IBD), and other inflammatory disorders [61]. The anti-inflammatory effects of saponins primarily hinge on the mechanisms shown in Figure 5.

Various saponins serve as adjuvants and immunostimulants, while also displaying hypocholesterolemic and antioxidant properties [4]. They enhance the immune response to infections and diseases. They can activate immune cells, such as macrophages and natural killer cells, which play a crucial role in combating infections. Additionally, saponins are used as adjuvants in vaccines to boost the immune response and improve vaccine efficacy.

Saponins possess antioxidant activity, helping to neutralize free radicals and oxidative stress in the body. Free radicals are unstable molecules that can damage cells and contribute to aging and various diseases. By scavenging these free radicals, saponins help protect cells from damage and promote overall health. Saponins have the capability to create non-soluble compounds with cholesterol, as well as other sterols and bile acids. They possess the ability to trap total cholesterol, LDL, and bile salts in the intestines, inhibiting their absorption, while not affecting HDL levels [62]. This leads to a reduction in blood cholesterol levels, specifically low-density lipoprotein (LDL) cholesterol, which is considered “bad” cholesterol. By lowering LDL cholesterol, saponins contribute to cardiovascular health and reduce the risk of heart disease.

Saponins exhibit antimicrobial effects against a wide range of pathogens, including bacteria, viruses, fungi, and protozoa. They disrupt microbial cell membranes and interfere with their replication, making them potential candidates for developing new antimicrobial agents and improving existing treatments. Oleanolic acid, derived from the root bark of *Newbouldia laevis*, exhibited wide-ranging antimicrobial properties when tested against six Gram-positive, twelve Gram-negative bacterial species, and three Candida species [63]. In animal nutrition, saponins are employed to decrease ammonia concentration and eliminate fecal odors [64]. These compounds are known for their antiprotozoal effects, achieved by forming complexes with the cholesterol present in protozoal cell membranes, leading to cell lysis and subsequent death. This antiprotozoal activity is particularly useful in reducing the populations of protozoa in the rumen of animals, thereby contributing to improved animal health and nutrition [27].

### 6.3. Flavor Characteristics of Saponins and Their Effect on the Flavor of Food Ingredients and Foods

Saponins are generally associated with a bitter taste; however, the structural and molecular chemical factors that govern the activation of the bitter taste response remain unknown. The bitterness can influence the overall flavor profile of foods containing these compounds. The bitterness of saponins may be more pronounced in certain types of foods and is correlated with the saponin concentration [65]. Soybean saponins were also found to have a bitter, astringent, and metallic flavor, and the presence of saponins in air-classified protein fractions could cause undesirable flavors [18,66]. The genetic varieties of plants such as pea seeds play key roles in determining saponin and resulting flavor profiles, as shown by Heng et al. [65].

Due to their complex molecular structures, saponins may interact with other flavor compounds, potentially altering the perceived taste of the final product. In some instances, the true food flavor may be masked by saponins, whereas in other instances foods may have unique interactions with saponins, leading to diverse flavor outcomes [67].

Although few studies have investigated the flavor attributes of saponins, research in this domain is not as extensive as in other areas of saponin bioactivities. Furthermore, individual sensitivity to bitter tastes can differ, leading to subjective perceptions of saponin flavors among individuals [68].

## 7. Applications of Saponins

Plant extracts containing saponins have been widely used in food and other industrial applications, mainly as surface active and foaming agents for centuries [69]. Recently, they have been regaining popularity, especially in skincare and cosmetics applications [70].

Among various plants, *Quillaja saponaria* extracts have been used as foaming agents in carbonated beverages and cosmetics, as emulsifiers in preparations containing lipophilic colors or flavors, and as preservatives [4,21]. Likewise, liquorice saponin extracts are used as flavor modifiers in baked foods, chewing gums, beverages, candies, herbs, seasonings, and dietary supplements [4].

Saponins in foods have traditionally been considered antinutritional factors [71], and in some cases their use has been limited due to bitter taste [72]. Therefore, most of the earlier research on food processing targeted the removal of saponins so that foods were as devoid of saponins as possible [72]. However, food and non-food sources of saponins have come into renewed focus in recent years due to increasing evidence of their health benefits, including their cholesterol-lowering ability, anti-inflammatory, immunostimulant, hypoglycaemic, antifungal, cytotoxic, and anticancer properties [62,73]. Recent research has established saponins as the active components in many herbal medicines [74,75], and highlighted their contributions to the health benefits of foods such as soybeans [76,77] and garlic [78]. The commercial potential of saponins has resulted in the development of new processes/processing strategies, and re-evaluation of existing technologies [79] for their extraction and concentration [80].

The ensuing sections will elaborate some of the applications of saponins.

### 7.1. Saponins as Natural Surfactants and Emulsifiers

Saponins, due to the presence of a lipid-soluble aglycone and water-soluble sugar chain, show amphiphilic characteristics. This structural make-up gives saponins a surface-active propensity similar to that of soaps or detergents. With one hydrophilic component and one lipophilic component, when dissolved in water, saponins tend to align themselves with the lipophilic part away from water, which leads to a reduction in the surface tension and causes foaming [81]. It is well understood that when the concentration of saponins is above the critical micelle concentration (CMC), they are able to form micelles (as shown in Figure 6) in aqueous solution. Consequently, saponins can enhance the solubility of other substances. Compared to synthetic surfactants, saponins are more effective in enhancing polycyclic aromatic hydrocarbons’ solubilization [82].

The size and structure of micelles are dependent on the structure of saponins. For example, saponins from *S. officinalis* and soybean bean form small micelles consisting of only two molecules, while the aggregates of *Quillaja saponaria* saponins consist of 50 molecules. It was documented that the properties and the aggregation number (number of monomers) of micelles formed by quillaja saponins are affected by temperature, salt concentration, and pH level. Saponins from *Quillaja saponaria* have a CMC between 0.5 and 0.8 g/L at 25 °C, and the CMC decreases with increasing salt concentrations [83]. The micelle shapes depend on the saponin molecules. For example, micelles formed by saponins from *S. officinalis* and *Quillaja saponaria* are elongated or even filamentous, while those formed by saponins of *Glycine max* are rather circular. It is thought that the reason for these differences is the chemical structure of the aglycone.

**Figure 6 ijms-24-13538-f006:**
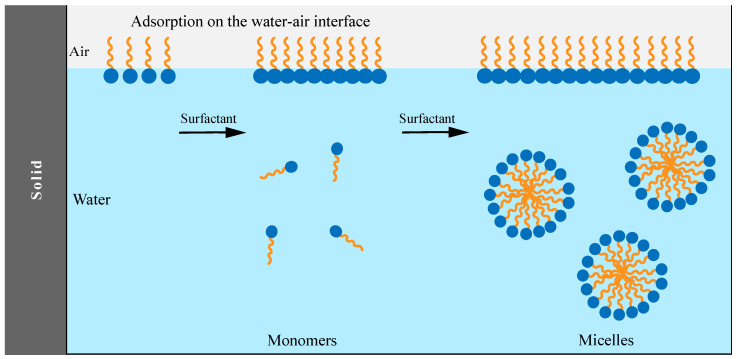
Schematic diagram of micelle formation (adapted from [84]).

The presence of carboxylic acid in the saponin molecular structure may strongly influence its surface activity. The location of this acid in the molecule has a particular importance. For example, Soybean saponins contain a carboxyl group in their hydrophilic part. The carboxyl group dissociates in the aqueous phase and forms free carboxyl anions, which are responsible for increasing the solubility of saponins in water environments. In contrast, saponins of Indian soapberry/washnut/ritha (*Sapindus mukorossi*) also contain the carboxylic groups, but they are attached to the hydrophobic aglycone. Consequently, the dissociation level of -COOH groups is very low. Saponins can also form mixed ‘sandwich-like’ or ‘pile of coins-like’ micelles with bile acids [83]. These are much larger than the micelles of saponins alone, and they differ depending on the structure of the aglycone. In the presence of bile acids, saponins from *Saponaria officinalis* and *Quillaja saponaria* form filamentous structures, while *Glycine max* saponins have an open structure. The ability of saponins to form large stable micelles with bile acids has important implications for dietary mechanisms. Saponins in food and feed increase fecal excretion of bile acids [83]. Additionally, the incorporation of cholesterol into saponin micelles increases their size, CMC, viscosity, and the aggregation level, resulting in the enhanced solubility of cholesterol. The micelles formed are too large for the digestive tract to absorb. This mechanism leads to a decrease in the plasma cholesterol concentration. Saponin *Quillaja saponaria* was found to solubilize cholesterol significantly better than linear hydrocarbon chain surfactants [85].

Interaction between saponins and membrane-bound cholesterol leads to pore formation and increasing of membrane permeability. This specific effect of saponins depends on a combination of factors, including the membrane composition, the type of saponin, and especially the nature of aglycone [86]. Saponins also form complexes with sterols in mucosal cell membranes, resulting in an increase in the intestinal mucosal cells’ permeability. Thus, this facilitates the uptake of substances to which the gut would normally be impermeable, for example, α-lactoglobulin [83].

Emulsifiers play two key roles in the creation of successful emulsion-based products. They facilitate the initial formation of fine lipid droplets during homogenization and enhance the stability of the lipid droplets once they have been formed. Oil-in-water emulsions may be formed using either high- or low-energy approaches. High-energy approaches utilize mechanical devices such as homogenizers, microfluidizers, high shear mixers, colloid mills, and sonicators. Quillaja saponin is a natural effective emulsifier to form and stabilize oil/water emulsions with very small oil beads (d  <  200 nm). They are stable within a wide range of environmental parameters (pH, ionic strength, temperature). This fact makes quillaja saponins suitable for wide application in food products [87]. Quillaja saponins currently find commercial applications as emulsifiers with milk and egg proteins such as β-lactoglobulin, β-casein or egg lysozyme via stabilization through electrostatic and hydrophobic interactions as well as via specific sugar binding sites [88].

One investigation into the emulsifying properties of quillaja saponins was carried out by Pekdemir et al. [89], who screened for a natural surfactant to be used for the emulsification of Ekofisk crude oil from the North Sea. The researchers found that quillaja bark saponin was able to emulsify the crude oil even at low concentrations of 0.1%, albeit only to a limited oil-to-surfactant ratio [89]. In recent years though, many studies have been performed to elucidate the formation of emulsions stabilized by various quillaja saponin products and manufactured with different homogenization techniques [90]. The observed emulsion-stabilizing properties were attributed to a strong electrostatic repulsion provided by quillaja saponins, because they have an unusually high negative ζ-potential of approximately −60 mV between pH 3 and 9 [91,92]. An additional contributor is their fast adsorption kinetics [92,93]. Lower mean droplet sizes of emulsions stabilized with quillaja saponins can be achieved at higher saponin concentrations [91,94], at high homogenization pressures [91], and after several homogenization passes [91,92,95].

### 7.2. Saponins as Natural Foaming Agents

A foam consists of a gas dispersed in a liquid, solid, or gelled matrix in the form of bubbles [96]. Basically, the gas bubbles (often air) are surrounded by a continuous thin liquid film, the so-called lamella, and thin film intersections (plateau borders) (Figure 7), forming interstitial spaces between the bubbles, and thus creating a three-dimensional network [97]. The formation of foams requires energy and is facilitated by whipping, shaking, or sparging of a solution [96]. Depending on the chosen application, the continuous liquid phase can additionally be gelled or solidified after the foam has been generated [98]. Foams are thermodynamically unstable, and as such are prone to different instability mechanisms induced by gravitational and van der Waals forces, leading to drainage, coalescence, and coarsening [99]. Therefore, their stabilization requires surface-active compounds such as surfactants and (bio)polymers and particles such as silica and polystyrene latex particles [99]. These surface-active molecules adsorb to the gas–liquid interface and reduce the interfacial tension, thus enabling the formation and stabilization of foams. The stabilization mechanisms and kinetic stability of the foams depend on the characteristics of the surface-active compounds used [99].

Generally, saponins with one sugar chain have superior foaming characteristics compared to those containing more sugar chains [83]. Quillaja saponins are well known for their foaming ability. In fact, the indigenous peoples of Chile used the aqueous solution of *Quillaja saponaria* bark to wash their hair and clothes as it produces a foam-like soap lather [4]. The quality and quantity of foam produced in the extract is used to qualitatively measure the concentration of saponins in the extracts [69]. Quillaja extract exhibits a good foam-stabilizing ability, with 85% of the foam still intact after 1 h of storage [100]. Foams stabilized with quillaja were found to be more stable at a lower pH (pH 3) and higher ionic strength (500 mM NaCl). It is suggested that bidesmosidic nature of quillaja saponins helps in reducing the destabilization of the membrane in foams [100].

A study involving the foaming attributes of the saponins from *Camellia oleifera* showed that the crude saponin content in the defatted seed meal of *C. oleifera* was 8.34%, and the total saponins content in the crude saponins extract was 39.5% (*w*/*w*) [101]. The foaming power of the 0.5% crude saponins extract solution from defatted seed meal of *C. oleifera* was 37.1% compared to that of 0.5% sodium lauryl sulfate or Tween 80 solutions.

The green fruits of yerba mate (*Ilex paraguariensis*), a South American plant, are a rich source of non-toxic and very low haemolytic saponins [102]. A study conducted to compare the effectiveness of mate saponin fraction (MSF) with sodium dodecyl sulfate (ionic surfactant) and polysorbate 80 (non-ionic surfactant) showed that the foamability of MSF and both reference surfactants were equivalent. The addition of MgCl_2_ resulted in a negative effect on MSF foamability. The salts NaCl, KBr, and KNO_3_ exhibited a negative influence on MSF foam lifetime and film drainage.

### 7.3. Saponins as Natural Antioxidants

Saponins are well recognized for their antioxidant activities. A higher free radical scavenging capacity was found for quillaja saponin extract compared to lecithin when using an oxygen radical absorbance capacity assay [103]. Quillaja saponin extract was also reported for its ability to cause a significant reduction in hydroperoxide and propanal (propionaldehyde) formation in nanoemulsions stabilized by saponin-rich extract compared to lecithin, SDS and Tween 80-based systems [103]. Ivy leaf extract is a rich source of triterpenoid monodesmosidic saponins, which exhibit a high antioxidant activity, DPPH radical and superoxide anion scavenging, hydrogen peroxide scavenging and metal chelating activities [104]. These saponins demonstrate expectorant, mucolytic, spasmolytic, bronchodilatory, and antibacterial effects, and are widely used in the treatment of bronchitis and pneumonia [105]. A study that investigated the possible antiradical and antioxidant activity of the monodesmosidic and crude extract of *Leontice smirnowii* showed a strong inhibition effect of peroxidation of linoleic acid emulsion [106]. It has been reported that in some legume saponins such as those from soybeans, kidney beans, peanuts, chickpeas and clover, antioxidant properties are associated with the presence of 2,3-dihydro-2,5-dihydroxy-6-methyl-4H-pyran-4-one (DDMP) linked to the C-22 of saponin aglycones [107,108]. In some saponin extracts including those from quillaja, antioxidant properties are associated with the phenolic compounds and their presence at the interface, facilitated by saponin molecules [109].

Han et al. [110] investigated the contents of saponins and phenolic compounds in relation to their antioxidant activity, as well as the α-glucosidase inhibition activity of several colored quinoa varieties. It was found that a higher degree of milling or polishing (i.e., removal of the outer layer) can reduce the contents of saponins, total phenolics, and anti-nutritional factors and improve their sensory quality, irrespective of varietal differences. Saponins and phenolic compounds significantly contribute to the antioxidant activities of quinoa. In another study, quinoa sprouts showed better antioxidant activity than fully grown parts of the quinoa plant. Overall, root and sprout had a higher antioxidant capacity compared to other parts of the quinoa plant, suggesting the potential use of quinoa root and sprout as a nutraceutical ingredient in the health food industry [111].

A study of the antioxidant activities of *Aralia taibaiensis*, a natural medicinal and food plant that is rich in triterpenoid saponins, in D-galactose-induced aging rats showed that it possesses a radical scavenging effect and can alleviate D-gal-induced aging damage in rats [112]. The saponins from *Hedera helix*, and *Hedera colchica* exhibited a strong total antioxidant activity. Four different saponins (α-Hederin, hederasaponin-C, hederacolchisides-E and -F) isolated from the leaves of *Hedrea helix* were evaluated for their ability to inhibit lipid oxidation. At the concentration of 75 pg/mL, these saponins (α-Hederin, hederasaponin-C, hederacolchisides-E and -F) showed 94, 86, 88, and 75% inhibition on lipid peroxidation of linoleic acid emulsion, respectively. These various antioxidant activities were compared with model antioxidants such as α-tocopherol, butylated hydroxyanisole (BHA), and butylated hydroxytoluene (BHT) [104]. Inhibition of only 65% was shown by α-tocopherol, whereas BHA and BHT showed 90% and 95% inhibition of lipid oxidation [104]. This indicated that saponins are superior to α-tocopherol and comparable to synthetic antioxidants BHA and BHT in inhibiting peroxidation.

Among the crude and total saponin fractions of *Chlorophytum borivilianum*, the crude extract showed higher free radical scavenging activity (2578 ± 111 mg ascorbic acid equivalents/100 g) and bleaching activity (IC_50_ = 0.7 mg mL^−1^), whist the purified saponin fraction displayed higher ferrous ion-chelating capacity (EC_50_ = 1 mg mL^−1^) [113].

### 7.4. Medicinal Applications of Saponins

Saponins are considered pro-drugs as they are converted to pharmacologically active substances after metabolization in the body [114]. Various in vivo studies have established their hemolytic [115], anti-inflammatory [116], antibacterial [117], antifungal [118], antiviral [119], insecticidal [120], anticancer [121], cytotoxic [122], hepatoprotective and molusccidal [123] properties. In addition, saponins are reported to exhibit cholesterol-lowering action in animals and humans [124,125] and have been found effective in decreasing blood glucose levels in diabetic patients [126]. Several mechanisms have been proposed to explain the hypocholesterolemic activity of saponins. Possible mechanisms may involve the capacity of saponins to form insoluble complexes with cholesterol, interfere with bile acid metabolism, and inhibit lipase activity, or regulate cholesterol homeostasis via monitoring the expression of the key regulatory genes of proteins or enzymes related to cholesterol metabolism [127,128]. The cholesterol-lowering activity of saponins has been demonstrated in both animal and human trials. Animal diets containing purified saponins or concentrated saponin extracts containing digitonin (saponin from *Digitalis purpurea*), saikosaponin (saponin from *Bupleurumfalcatum* and related plants), as well as saponins from saponaria, soybean, chickpea, yucca, alfalfa, fenugreek, quillaja, gypsohila, and garlic resulted in reductions in cholesterol concentrations [4].

Saponins can also be beneficial for hyperlipidaemia and are capable of reducing the risk of heart disease in humans [3]. Saponins may play a major role in protection from cancer. Research on colon cancer cells suggests that it is the lipophilic saponin cores that may be responsible for this biological activity [3]. A study of the relationship between the chemical structure of aglycones and the colon anticancer activity of saponins revealed that sapogenols were more bioactive than glycosidic saponins. Other aglycones with anticancer activity include dammarane sapogenins from ginseng, betulinic acid, and oleanolic acid. These compounds were also reported to possess anti-viral, anti-inflammatory, hepatoprotective, anti-ulcer, antibacterial, hypoglycaemic, anti-fertility, and anticariogenic activities. However, the conversion of saponins to their aglycones may result in the loss of activity [129]. For example, the hydrolysis of saponins by ruminal bacteria results in the loss of antiprotozoal activity. Similarly, the deacylation of quillaja saponins decreases their adjuvant activity [130].

Due to the structural complexity and toxicity of plant saponins, their use in human vaccines is limited, but progress in new processing and purification techniques that maintain immunological adjuvant activity is important to create saponins as new-generation vaccines [2].

A steroidal saponin glycoside isolated from *Fagonia indica* was found to induce cell-selective apoptosis or necrosis in cancer cells. The clinical significance of triterpenoid saponins in the prevention and treatment of metabolic and vascular disease is noteworthy [114].

Saponins from various sources are important constituents of traditional folk medicines. Ginsenosides are saponins produced by Panax species which are known for their antioxidant, anti-inflammatory, and anti-cancer activities [131]. It has been reported that most saponins form insoluble complexes with 3-β-hydroxysteroids and are known to interact with bile acids and cholesterol, forming large mixed micelles. These functionalities are thought to result in the cholesterol-lowering capacities of saponins in some animal species; however, their hypocholesterolemic effects in humans are more speculative [3].

Although saponins are considered beneficial in several medical conditions and are being used as alternative medical substances, a detailed understanding of the relationship between the chemistry of saponins and their interactions with signaling and other biological pathways and systems is necessary to confirm their actions and safety for human or animal use. Multidisciplinary approaches involving chemists, physicians, toxicologists, molecular biologists, and others will be essential to explore and define the potential of saponins in this field [132].

## 8. Bioavailability of Saponins

Saponins are generally regarded as having low bioavailability. The absorption of saponins in the human diet is highly variable, and is affected by several factors, including the amount of saponins consumed in a meal, interaction of saponins with bile acids and other micronutrients, food processing methods, and metabolic adaptation of individuals to dietary saponins [128]. Saponins impart a bitter taste in dietary plants at high concentrations, which ultimately reduces the consumption of saponins by animals and humans [133]. It has also been reported that acetyl–soybean saponins taste more bitter than nonacetylated constituents [134].

Studies involving the physiological digestion and absorption of saponins in the human body have revealed that they have a longer residence time in the gastrointestinal tract and are poorly absorbed. This is mainly attributed to their large molecular mass (>500 Da), high hydrogen-bonding capacity (>12), and high molecular flexibility (>10), resulting in poor membrane permeability [135]. Various in vivo studies with rats, mice, and rabbits have demonstrated that saponins are not absorbed in the alimentary canal, and largely pass to the large intestine where they are hydrolyzed enzymatically to aglycones, known as sapogenins, and sugars [4]. Sapogenins derived in the gut via microbial biotransformation from saponins typically have higher lipid solubility and are more readily absorbed in the body [136].

The interaction of saponins with minerals such as zinc and iron and phytate–mineral complexes results in a reduction in the bioavailability of both the saponin and the minerals [8]. Saponins from alfalfa and soybean were reported to decrease iron and zinc absorption in rats and pigs [8].

## 9. Food Regulations on Saponin Products

It is important to note that the use of saponins in food applications requires careful consideration of their concentration, interactions with other food components, and potential effects on taste and texture. Additionally, specific regulations and safety guidelines regarding the use of saponins in food products may vary across different countries or regions.

Food Standards Australia and New Zealand approved the use of saponin-rich quillaja/quillaja extract as a Food Additive (Emulsifier) in 2013. Quillaia extracts type 1 (INS 999i), and type 2 (INS 999ii) are listed in the Codex Alimentarius General Standard for Food Additives. The permissions are for addition to specific types of water-based flavored drinks, including “sport”, “energy”, or “electrolyte” drinks and particulate drinks. The maximum level of addition is 50 mg/kg, expressed on a saponins basis, but only for quillaja extract type 1. Their functional class is listed as emulsifier and foaming agent. Both extracts differ in their purity and the concentrations of the active ingredients, i.e., saponin. Similarly, the European Food Safety Authority has also permitted quillaja extract (E999) for use in non-alcoholic flavored beverages and cider (excluding cidre bouché), to a maximum level of 200 mg/L, as an anhydrous extract.

Quillaja extract has generally recognized as safe (GRAS) status in the USA, for its use as an emulsifier or encapsulating agent in beverage products to deliver fats, nutrients, vitamins, colors, and clouding agents to a similar range of beverages, and also as a foaming agent for semi-frozen carbonated and non-carbonated beverages. It is also permitted as a flavoring adjuvant, with technological functions as an emulsifier, stabilizer, or foam stabilizer for both natural and synthetic flavors. Similarly, quillaja extract has been approved in Canada as a miscellaneous food additive in beverage bases, beverage mixes, and soft drinks as a foaming agent. Saponin-rich quillaia extract is permitted in several other countries (China, Japan, India, Singapore, Thailand, Taiwan, and Vietnam) in food and pharmaceutical applications. The existing permissions are for its use with flavors, as an emulsifier or stabilizer, or as a foaming agent for a range of beverages at a wide range of levels from 50 mg/kg saponins to 1500 mg/kg. As such, regulatory advice will need to be sought prior to introduction of any saponin-containing extracts or ingredients to the Australian foods and beverages or supplements sector, as indeed is the case for their introduction to markets in other jurisdictions.

## 10. Concluding Remarks and Future Perspectives

In conclusion, saponins are a diverse group of compounds with unique structural and chemical properties and abundant occurrence in the plant kingdom. Their structural diversity, characterized by hydrophilic sugar moieties attached to lipophilic aglycones, contributes to their wide-ranging functional and biological properties. Due to increased consumer focus on natural ingredients, saponins’ application in the food, cosmetic, and pharmaceutical sectors could witness a substantial growth in the coming decade, if structure–function relationships can be further understood and tailored to end-use.

In food applications, saponins serve as natural emulsifiers, foaming agents and stabilizers, resulting in enhanced texture and stability in food products. They also exhibit potential health benefits, including cholesterol-lowering and anticancer effects, making them attractive functional food ingredients. Saponins derived from certain plants have shown promising health benefits, such as cholesterol-lowering or immune system modulatory effects. Similarly, saponins’ antimicrobial properties could be explored for their potential as natural preservatives in food products, as they may help to inhibit the growth of spoilage-causing microorganisms and extend the shelf life of perishable goods. They may also be incorporated into (edible) packaging as antimicrobial agents, and help address the challenge of extending the shelf-life of foods. However, barriers such as bitterness and cytotoxicity need to be addressed to fully exploit the potential of saponins in the food industry.

In the pharmaceutical field, saponins possess diverse bioactivities, including anti-inflammatory, antimicrobial, antiviral, and antiparasitic properties. Their cytotoxic activity against cancer cell lines has also generated interest in their potential as anticancer agents. Furthermore, saponins can act as adjuvants, improving immune response to vaccines, and as drug delivery agents, enhancing solubility and bioavailability. In the future, they may be employed as excipients or carriers in pharmaceutical formulations to improve drug delivery and increase the efficacy of therapeutic compounds. Some saponins have exhibited cytotoxic properties against cancer cells. Further research may focus on the development of saponin-based drugs or formulations for targeted cancer therapies, potentially complementing existing treatment modalities. As for applications in cosmetics, saponins have cleansing and foaming properties, making them potential alternatives to synthetic surfactants in personal care products such as shampoos, soaps, and body washes. Their natural origin and mildness may appeal to consumers seeking sustainable and skin-friendly options. Due to their reported antioxidant and skin-soothing properties, saponins may find applications in cosmetics and skincare products. They could be explored as natural ingredients in formulations for anti-aging, skin hydration, or soothing sensitive skin.

Future research on saponins might focus on addressing the challenges associated with their utilization, such as bitterness and cytotoxicity. Further investigations into their mechanisms of action, bioavailability, and interactions with other compounds in the human body are needed to fully understand their potential for food, cosmetic, and pharmaceutical applications. Additionally, exploring novel sources of saponins and developing sustainable extraction methods can broaden their availability and enhance their commercial viability. In summary, saponins offer exciting prospects for the food and associated industries due to their unique chemistry, diverse functionalities, and potential health benefits. Continued research and innovation will pave the way for their successful integration into various applications, thereby contributing to improved human health and wellbeing.

## Figures and Tables

**Figure 1 ijms-24-13538-f001:**
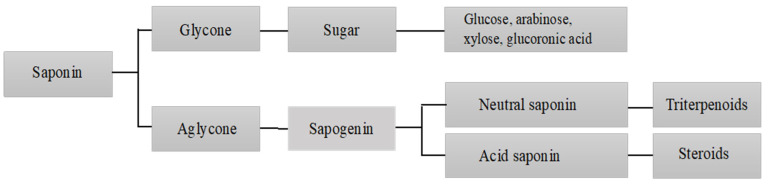
Schematic diagram of the chemical composition of saponins.

**Figure 2 ijms-24-13538-f002:**
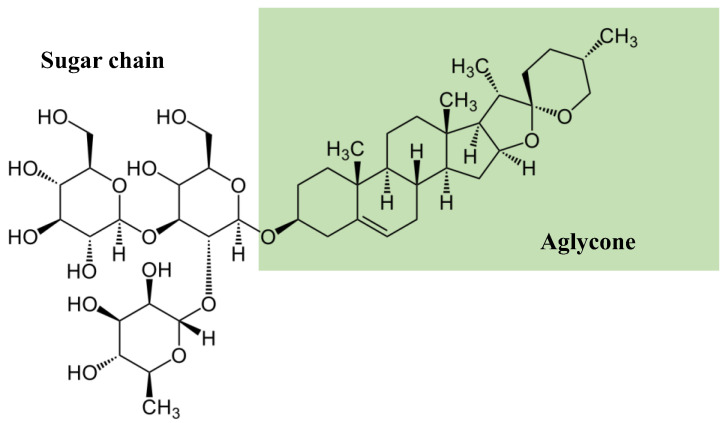
Structure of saponin (adapted from [27]).

**Figure 3 ijms-24-13538-f003:**
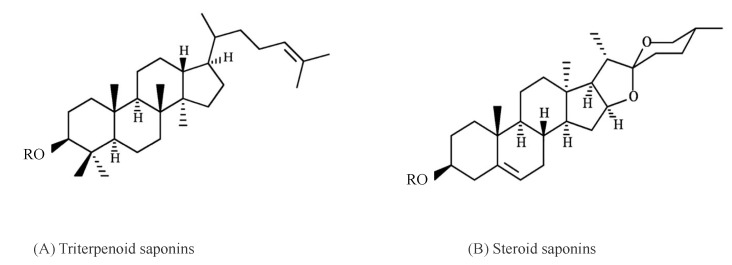
Basic skeletons of saponins. (**A**) Triterpenoid saponins and (**B**) steroid saponins (adapted from [4]).

**Figure 4 ijms-24-13538-f004:**
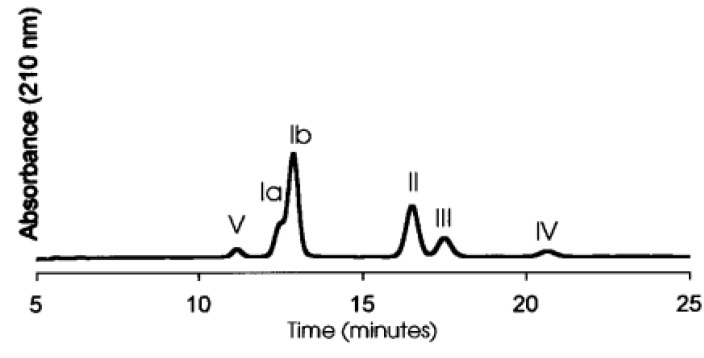
HPLC chromatogram of group B soyasaponin fractions (Reprinted from [39] with permission). Ia: Soyasaponin I, Ib: Soyasaponin I, II: Soyasaponin II, III: Soyasaponin III, IV: Soyasaponin IV, V: Soyasaponin V.

**Figure 5 ijms-24-13538-f005:**
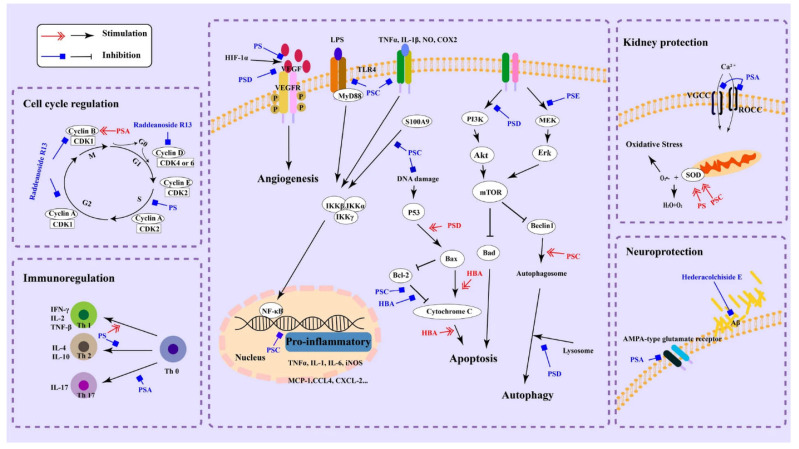
The molecular mechanisms for the pharmacological activities of pulsatilla saponins (reprinted with permission through the Creative Commons license from [60]).

**Figure 7 ijms-24-13538-f007:**
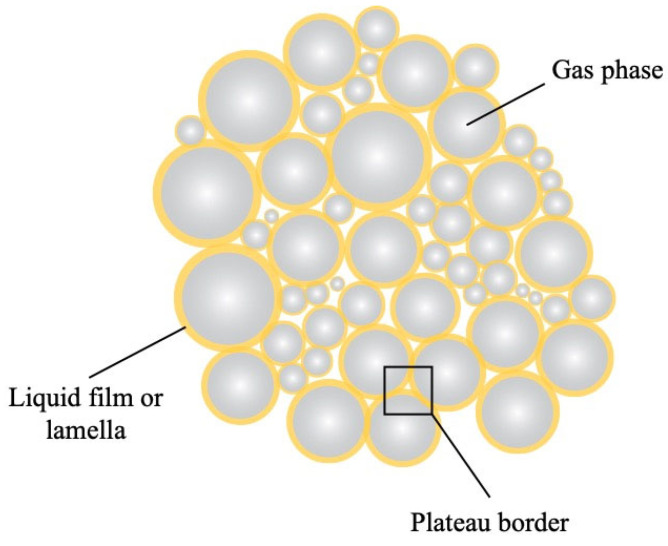
Schematic diagram of a foam structure (adapted from [97]).

**Table 1 ijms-24-13538-t001:** The better-known botanicals rich in saponins (adapted from [1,4,11,12,17]).

Latin Name	Common Name	Saponin Content [%]	Reference
*Aesculis hipocastanum*	Horse chestnut	3.0	[18]
*Asparagus officinalis*	Asparagus	1.5	[17]
*Avena sativa*	Oats	0.1–0.2	[18]
*Beta vulgaris*	Sugar beet (leaves)	5.8	[18]
*Chenopodium quinoa*	Quinoa	0.1–2.3	[1]
*Cicer arietinum*	Chickpea	0.2	[19]
*Crocus savitus*	Saffron crocus	1.2–3.4	[1]
*Glycine max*	Soybean	0.2–0.5	[19]
*Glycyrrhiza glabbra*	Liquorice	22.2–32.3	[19]
*Hedera helix*	Ivy	5.0	[1]
*Medicago sativa*	Alfalfa	0.1–1.7	[19]
*Panax ginseng*	Chinese ginseng	2.0–3.0	[1]
*Panax quinquefolius*	American ginseng	1.4–5.6	[20]
*Pisum sativum*	Green pea	0.2–4.2	[18]
*Polygala* spp.	Milkwort	8.0–10.0	[1]
*Primula* spp.	Primula	5.0–10.0	[1]
*Quillaja saponaria*	Quillaja bark	9.0–10.0	[21]
*Saponaria officinalis*	Soapwort	2.0–5.0	[1]
*Smilax officinalis*	Sarsaparilla	1.8–2.4	[1]
*Trigonellafoenum-graecum*	Fenugreek	4.0–6.0	[22]
*Vicia faba*	Faba beans	0.35	[17]
*Yucca schidigera*	Yucca	10.0	[23]

**Table 2 ijms-24-13538-t002:** Extraction techniques for saponins (adapted from [11,29,30]).

Extraction Method	Solvent System	Temperature	Extraction Time	Remarks
Simple extraction	Water or 50–98% MeOH/EtOH	Varies between 50–90 °C	Hours to days	Low yield. Long extraction time. High solvent use
Soxhlet Extraction	50–98% ethanol	Up to 80 °C	24–72 h	Thermal degradation of labile components
Reflux Extraction	50–98% ethanol	Up to 80 °C	1–4 h	Thermal degradation of labile components
Ultrasound-assisted extraction	Water, MeOH/EtOH	0–90 kPa (Acoustic)	0.5–6 h	Higher saponin yield. Shorter extractiontime. Less solvent use
Microwave-assisted extraction	Water, MeOH/EtOH		~5 min	Higher saponin yield. Shorter extractiontime. Less solvent use
Pressurized solvent extraction	Water, MeOH/EtOH	100 °C at 150 psi	0.5–6 h	Higher saponin yield. Shorter extraction time. Less solvent use
Supercritical fluid extraction	CO_2_ + EtOH	25 MPa 35–55 °C	1–8 h	Sequential extraction with increasing polarity with co-solvent

**Table 3 ijms-24-13538-t003:** Various methods of quantification of saponins.

Analytical Method	Sensitivity	Selectivity	Accuracy	References
HPLC	High sensitivity: detects in ppm to ppb range	High selectivity due to chromatographic separation. Detects multiple components in complex mixtures	Accurate quantification with proper calibration and validation	[46,50]
HPTLC	Moderate to high sensitivity: detects in mg/mL range	-	Accurate quantification with proper calibration and validation	[50,51]
UPLC	High sensitivity: detects in ppm range Higher pressures result in more efficient separation	Improved selectivity due to efficient separation in a short time	Accurate quantification with proper calibration and validation	[42,52]
LC/MS	Very high sensitivity: detects in ppb to ppt range	MS offers higher selectivity through accurate mass detection. Tandem MS enhances selectivity	Accurate quantification with proper calibration and validation. Provides compound identification	[47,48]
UV-Vis Spectroscopy	Moderate sensitivity: detects in mg/mL range	Limited selectivity, and may be affected by interferences from other compounds with similar absorbance spectra	Accuracy depends on sample purity and wavelength selection and calibration	[31,53,54]
FTIR Spectroscopy	Moderate sensitivity: detects in mg/mL range	Limited selectivity due to overlapping bands in IR spectra of different compounds. IR spectra can provide functional group information	Accuracy is affected by sample preparation, baseline correction and calibration	[55,56]

## Data Availability

No new data were created or analyzed in this literature review. Data sharing is not applicable to this article.

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
