# Peer review of "Perspectives on Saponins: Food Functionality and Applications"

_ijms, 2023, doi:10.3390/ijms241713538_

Round 1

Reviewer 1 Report

Please, see the file attached

The quality of English is good, check only for minor spelling errors and some sentence which is not clear.

Author Response

We thank the Section Managing Editor for prompt reviews and providing us with the opportunity to revise the manuscript. We also like to thank all the reviewers for their constructive feedback. We have now addressed all the comments from the reviewers and revised the manuscript accordingly. The changes made as part of this revision are written in the red font. We believe that the science and the presentation of the manuscript have improved substantially after this revision.

Reviewer 2 Report

The paper is well written. The authors did a good job of describing the biological effects of saponin, and I really appreciate that they mention the legal concerns. The isolation and analysis procedures in the paper are where I have the most criticism. I believe that we did not get the same high-quality overview of previous research as in the rest of the paper. My recommendation is to either enlarge that section and add the section regarding the difficulties with saponin analysis or to completely omit that section from the manuscript.

More detail could be included to the paragraph on the mass spectrometric analysis of saponins. Since it is challenging to identify saponins without fragmentation, the majority of the publications you listed used tandem MS (from the paper, it is unclear whether just MS or MSMS was used). I suggest providing a few additional examples of mass spectrometer setups that were applied to analysis along with a brief overview of the difficulties with LC-MSMS analysis.

L177-L188 In my opinion, it is not necessary to go into great detail on the methods in the review paper.

Some other minor comments:

It is clear that Table 1 is taken from another paper. However, it would be beneficial to provide citations for the studies on each saponin-rich plant mentioned (works in which saponin content in plants has been confirmed).

Please make sure that the lines in picture 1 don't cross over the letters.

L126 - missing references

L135 - It would be good to put a reference to claim the most commonly used solvents.

L147 - missing space between the words various and chromatography

Author Response

(The authors gave the same response as above.)

Round 2

Reviewer 1 Report

I appreciate the work made by authors to fulfill my comments and I think the manuscript have been substantially improved. The paragraphs relative to extraction and quantification are nice and tables are very useful. I have only some minor issues to point out:

Line 151: the website is not available, maybe there is a reference about this polymer to report?

Line 192-194: check the grammar of this sentence

The quality of English language is good
